# Emerging *Cryptococcus gattii* species complex infections in Guangxi, southern China

**Chunyang Huang**[1]☯, **Clement K. M. Tsui**[2,3,4]☯, **Min Chen**[5], **Kaisu Pan**[1], **Xiuying Li**[1], **Linqi Wang**[6], **Meini Chen**[7], **Yanqing Zheng**[1], **Dongyan Zheng**[1], **Xingchun Chen**[8], **Li Jiang**[1], **Lili Wei**[1], **Wanqing Liao**[5]*, **Cunwei Cao**[1]*

**1** Department of Dermatology and Venereology, The First Affiliated Hospital of Guangxi Medical University, Nanning, P. R. China, **2** Department of Pathology, Sidra Medicine, Qatar, **3** Department of Pathology and Laboratory Medicine, Weill Cornell Medicine–Qatar, Doha, Qatar, **4** Division of Infectious Diseases, Faculty of Medicine, University of British Columbia, Vancouver, BC, Canada, **5** Shanghai Key Laboratory of Molecular Medical Mycology, Department of Dermatology, Changzheng Hospital, Second Military Medical University, Shanghai, P. R. China, **6** State Key Laboratory of Mycology, Institute of Microbiology, Chinese Academy of Sciences, Beijing, P. R. China, **7** Clinical Medicine (8-year program), XiangYa School of Medicine, Central South University, Changsha, P. R. China, **8** The People's Hospital of Guangxi Zhuang Autonomous Region, Nanning, P. R. China

☯ These authors contributed equally to this work.
* dumingwei7939@126.com (WL); caocunwei@yeah.net (CC)

**Data Availability Statement:** MLST nucleotide sequences for the eleven clinical isolates determined in this study have been deposited in the

## Abstract

The emergence and spread of cryptococcosis caused by the *Cryptococcus gattii* species complex has become a major public concern worldwide. *C. deuterogattii* (VGIIa) outbreaks in the Pacific Northwest region demonstrate the expansion of this fungal infection to temperate climate regions. However, infections due to the *C. gattii* species complex in China have rarely been reported. In this study, we studied eleven clinical strains of the *C. gattii* species complex isolated from Guangxi, southern China. The genetic identity and variability of these isolates were analyzed via multi-locus sequence typing (MLST), and the phylogenetic relationships among these isolates and global isolates were evaluated. The mating type, physiological features and antifungal susceptibilities of these isolates were also characterized. Among the eleven isolates, six belonged to *C. deuterogattii*, while five belonged to *C. gattii sensu stricto*. The *C. deuterogattii* strains from Guangxi, southern China were genetically variable and clustered with different clinical isolates from Brazil. All strains were *MAT*α, and three *C. deuterogattii* isolates (GX0104, GX0105 and GX0147) were able to undergo sexual reproduction. Moreover, most strains had capsule and were capable of melanin production when compared to the outbreak strain from Canada. Most isolates were susceptible to antifungal drugs; yet one of eleven immunocompetent patients died of cryptococcal meningitis caused by *C. deuterogattii* (GX0147). Our study indicated that the highly pathogenic *C. deuterogattii* may be emerging in southern China, and effective nationwide surveillance of *C. gattii* species complex infection is necessary.

GenBank nucleotide sequence database under accession numbers MK344035-MK344111.

**Funding:** This study was supported by grants from the National Natural Science Foundation of China (Nos. 81571971 and 81271804) and the Natural Science Foundation of Guangxi Province of China (2017GXNSFAA198004, AB18221017 and 2018GXNSFAA294090). The funders had no role in study design, data collection and analysis, decision to publish or preparation of the manuscript.

**Competing interests:** The authors have declared that no competing interests exist.

## Author summary

Cryptococcosis is a fatal systemic fungal disease caused by *Cryptococcus neoformans/gattii* species complexes. As a former member of the *C. neoformans*, *C. gattii* had been easily neglected before being elevated to species level. Human *C. gattii* species complex infection was previously confined to the tropical and subtropical regions worldwide. However, in 1999, an outbreak of *C. gattii* species complex occurred on Vancouver Island in Canada then expanded to the Pacific Northwest in the USA, causing over 200 infections. The highly virulent, highly pathogenic and more resistant to antifungal drugs of this species have become a therapeutic problem. To initiate a better understanding of the infection characteristics and pathogenicity of *C. gattii* species complex in Guangxi, southern China, the current study aimed to characterize the *C. gattii* species complex isolates genetically and phenotypically. The ISHAM consensus MLST scheme was utilized to investigate the genetic structure of *C. gattii* species complex and to correlate their geographic origin, clinical source, virulence factors and antifungal susceptibility. The authors expect that this work can support surveillance and encourage more research and public health initiatives to prevent and control the cryptococcosis cause by *C. gattii* species complex.

## Introduction

*Cryptococcus* is a life-threatening fungal pathogen of humans and animals[1]. The infection process of *Cryptococcus* is usually via the inhalation of airborne spores (or yeast cells) into the respiratory tract and their subsequent dissemination to the central nervous system, causing pulmonary cryptococcosis and cryptococcal meningoencephalitis[1, 2]. During the past two decades, considerable genetic heterogeneity has been demonstrated to occur in the *C. neoformans/gattii* species complexes by a plethora of molecular methods. Various molecular biological techniques have been used to study the epidemiology and population structure of the *Cryptococcus gattii/neoformans* species complexes, including random amplification of polymorphic DNA (RAPD) analysis, PCR fingerprinting, amplified fragment length polymorphism (AFLP) analysis, multilocus microsatellite typing (MLMT) analysis and multi-locus sequence typing (MLST) analysis[3]. Recently, next-generation sequencing (NGS) technology has been utilized to investigate the epidemiology of *C. gattii* isolates[4]. However, the identification of the species is commonly based on the MLST protocol standardized by the International Society for Human and Animal Mycology (ISHAM)[5]. The MLST scheme has become an important tool for the characterization of the population genetic structure of the *Cryptococcus* species. Since the taxonomy of the polyphyletic genus *Cryptococcus* has been thoroughly revised, two varieties of *C. neoformans* have been recognized as species: *C. neoformans* (formerly *C. neoformans* variety *grubii*) and *C. deneoformans* (formerly *C. neoformans* variety *neoformans*)[6–8]. However, the molecular types of *C. gattii* species complex have been elevated to the species level as *C. gattii sensu stricto* (AFLP4/VGI), *C. deuterogattii* (AFLP6/VGII), *C. bacillisporus* (AFLP5/VGIII), *C. tetragattii* (AFLP7/VGIV) and *C. decagattii* (AFLP10/VGIV)[7]. The *C. deuterogattii* subtype (AFLP6A/VGIIa, AFLP6B/VGIIb and AFLP6C/VGIIc) caused an outbreak in the Pacific Northwest (PNW) region of Canada and the United States[9, 10].

 Cryptococcosis is a global and invasive systematic mycosis caused by *C. gattii/neoformans* species complexes, leading to morbidity and mortality in both immunocompetent and immunocompromised individuals, such as those with acquired immune deficiency syndrome (AIDS) or those undergoing organ transplantation[11]. Infections due to *C. neoformans*

species complex occur worldwide while cryptococcosis caused by *C. gattii* species complex was traditionally considered an endemic disease, and associated with tropical and subtropical climates[11]. However, the outbreaks of *C. deuterogattii* in humans and a wide range of mammals in Vancouver Island and the PNW of the USA demonstrates that the fungus has adapted to environments beyond the endemic (sub)tropical regions[9, 10]. The source of infection is usually traced back to *Eucalyptus* and other species of trees, such as *Ficus* spp. and *Terminalia* spp. (almond) trees[12, 13].

In Asia, the first case of *C. gattii* species complex infection was reported by Wanqing Liao in 1980 by the strain formerly named as $S_{8012}$ and now identified as *C. gattii s.s.*[14]. *C. gattii s. s.* is the most frequently encountered species worldwide[3], and only a few sporadic cases of *C. deuterogattii* infection have been reported in Asian countries such as India, Thailand, and Malaysia[15]. In 2007, Okamoto *et al.* reported the first case of cryptococcosis caused by a highly virulent *C. deuterogattii* subgenotype (AFLP6A/VGIIa) in Japan[16]. However, to our knowledge, reports of *C. gattii* species complex infection in China are limited. A study in 2008 showed that nine of 115 (7.8%) clinical *Cryptococcus* isolates were members of the *C. gattii* species complex, consisting of eight strains of *C. gattii s.s.* and one strain of *C. deuterogattii*[17]. Several sporadic *C. deuterogattii* infections have also been reported in mainland China, mainly in southern China[18, 19]. However, the genetic identity and variability of these isolates and the phylogenetic relationships among these clinical isolates and global isolates have yet to be thoroughly investigated.

In this study, we described eleven cases of cryptococcosis caused by *C. gattii* species complex infections between 2014 and 2018 in Guangxi, southern China. We determined their genotypes by MLST and studied their phylogenetic relationships with the global strains. We also characterized the mating type, physiological characteristics, virulence factors and antifungal susceptibility of these clinical isolates.

## Materials and methods

### Ethics statement

This study was approved by the Medical Ethics Committee of First Affiliated Hospital of Guangxi Medical University. All participants were adults. The clinical data in this study was obtained with the written consent of the patient or the patient's family and data collected concerning them was anonymized.

### Selection of *C. gattii* species complex from clinical isolates

To study the epidemiological characteristics of cryptococcosis caused by *C. gattii* species complex in Guangxi, southern China, all the clinical strains of *Cryptococcus* spp. stored in the Fungal Diseases Survey Center of Guangxi, the First Affiliated Hospital of Guangxi Medical University Guangxi, southern China, have been evaluated. In total, one hundred and twenty *Cryptococcus* strains were isolated from patients with clinically confirmed cryptococcosis between 2014 and 2018. All but three were isolated from the patients hospitalized in First Affiliated Hospital of Guangxi Medical University, which is the largest tertiary care hospital in the Guangxi Autonomous Region in southern China and has 2750 beds. Two strains were from The Fourth People's Hospital of Nanning and one strain was from the People's Hospital of Guangxi Zhuang Autonomous Region. Among the 120 strains, most of them were from incident patients, except one (GX0049) was a relapse patient. Six strains isolated from the bronchoalveolar lavage fluid (BALF), three from lung tissue and two from skin tissue; the other strains isolated from CSF. The data of the patients was collected using a standardized form that was based entirely on the medical reports of each patient, including demographic

information (age and gender), domiciles, birth and development details, medical history, clinical manifestations, laboratory data, imaging changes, diagnoses, treatments, and prognoses.

All strains were subcultured on Sabouraud Dextrose Agar (SDA) plates and incubated at 25°C, and then transferred onto L-Canavanine-glycine-bromothymol blue (CGB) medium for three to seven days to differentiate *C. neoformans* from *C. gattii* species complex as previously described [20]. In the CGB medium, eleven isolates had a positive reaction and turned the medium blue, while the rest of isolates failed to produce a color change, which were further identified as *C. neoformans* var. *grubii* by matrix-assisted laser desorption/ionization time-of-flight mass spectrometry (MALDI-TOF MS). Among the eleven strains of *C. gattii* species complex, ten strains were isolated from CSF, and one strain was isolated from lung tissue.

Reference *Cryptococcus* strains R265 (AFLP6A/VGIIa) and H99 (AFLP1/VNI) were acquired from Shanghai Key Laboratory of Molecular Medical Mycology, Changzheng Hospital, Second Military Medical University, Shanghai, China. *MAT*a strains (AFLP2/VNIV, AFLP1/VNI and AFLP6/VGII) were obtained from State Key Laboratory of Mycology, Institute of Microbiology, Chinese Academy of Sciences, Beijing, China.

## MLST and phylogenetic analysis

Isolates were cultured on SDA for 72 h prior to DNA extraction. Genomic DNA was extracted with the NuClean Plant Genomic DNA Kit (CWBIO, Beijing, China). Based on the ISHAM consensus MLST scheme, seven loci including capsule polysaccharide (*CAP59*), glycerol 3-phosphate dehydrogenase, (*GPD1*), laccase (*LAC1*), the intergenic spacer (*IGS1*) region, phospholipase B1 (*PLB1*), superoxide dismutase (*SOD1*), and orotidine monophosphate pyrophosphorylase (*URA5*) genes were amplified and sequenced[5]. The alleles were analyzed, and the STs were determined based on the MLST database (http://mlst.mycologylab.org). The sequences of the seven MLST loci have been deposited in the MLST database and GenBank.

Phylogenetic analysis of the *C. gattii* species complex was performed based on the alignment of seven concatenated nucleotide sequences (*CAP59*, *GPD1*, *LAC1*, *IGS1*, *PLB1*, *SOD1* and *URA5*)[21]. The genetic relationships among these Chinese clinical strains and strains in different countries[21–23] were investigated by MEGA7[24]; a phylogenetic tree was generated using the maximum likelihood method with a bootstrap analysis using 1000 replicates [24–26]. Principal component analysis (PCA) was performed with the Adegenet 2.1.1 package for software R (version 3.4.4) to explore the genetic relationships and geographic patterns of the strains[27].

## Mating type determination and physiological analysis

Mating types were determined by using specific primers targeting the *MF*α and *MF*a pheromone genes as previously described[28]. Mating experiments were performed by pairing eleven clinical strains and two reference strains (*MAT*α; AFLP6A/VGIIa R265 and AFLP1/VNI H99) with three strains of the opposite mating type (*MAT*a; AFLP2/VNIV, AFLP1/VNI and AFLP6/VGII) on V8 medium at 25°C in darkness for two weeks. The experiment was carried out twice, and the formation of hyphae and sexual structures was observed and investigated under a microscope.

Since certain strains were able to mate *in vitro*, we also studied and compared their virulence factors to the standard reference strain (R265). Melanin production, and capsule formation were evaluated using slightly modified protocols that were published previously[29–31]. Visual analysis of melanin production was performed on caffeic acid agar; strains were grown on agar plates incubated at 30°C and 37°C for 72 h and observed the appearance of brown yeast colonies. Capsule formation was induced with RPMI-1640 medium for 72 h at 37°C and

5% $CO_2$. The capsule size of at least 100 cells was quantified by light microscopy using encapsulated to naked yeast size (cell wall to cell wall diameter) ratios.

## Antifungal susceptibility testing

Antifungal agents fluconazole (FLC), fluorocytosine (5FC), amphotericin B (AMB), itraconazole (ITC), voriconazole (VOR), posaconazole (POS) and isavuconazole (ISA) were used in the susceptibility tests. The broth microdilution method was performed following the CLSI M27-A3 guidelines[32] to assess the antifungal susceptibility of all clinical isolates *in vitro*. The concentration ranges were 0.125–64 µg/mL for FLC, 0.008–4 µg/mL for 5FC, 0.016–8 µg/mL for AMB and 0.002–1 µg/mL for ITC, VOR, POS and ISA. The MIC values were recorded after incubation at 37˚C for 72 h. The MIC of AMB was defined as the lowest concentration of drug showing no yeast growth, while the MICs for other antifungal agents were defined as low drug concentrations that caused a prominent reduction in growth ($\geq$50%) compared with the drug-free growth control. Since there was no established breakpoints standard of antifungal drugs for *Cryptococcus*, therefore, epidemiological cutoff values (ECVs) has been offered to determine whether a strain is wild type (*in vitro* susceptible) or non-wild type (*in vitro* resistant). Based on the previous recommendation[33–35], the ECVs of *C. gattii s.s.* for FLC were 8 µg/mL; 4 µg/mL for 5FC; 0.5 µg/mL for AMB, ITC, VOR and POS; and 0.25 µg/mL for ISA. The ECVs of *C. deuterogattii* were 32 µg/mL for FLC; 16 µg/mL for 5FC; 1 µg/mL for AMB; 0.5 µg/mL for ITC and POS; and 0.25 µg/mL for VOR and ISA[33–35]. *Candida parapsilosis* ATCC22019 was used for quality control strain.

## Statistical analysis

Statistical analysis was performed using SPSS 17.0 (SPSS Inc., Chicago, IL, USA). The *p* values of the relative capsule size data were calculated using one-way ANOVA statistical analysis. $p < 0.05$ was considered statistically significant.

## Accession numbers

MLST nucleotide sequences for the eleven clinical isolates determined in this study have been deposited in the GenBank nucleotide sequence database under accession numbers MK344035-MK344111.

# Results

## Clinical characteristics of *C. gattii* species complex cases

The clinical manifestations and medical histories of the 11 patients are summarized in Table 1. A total of ten cryptococcal meningitis (CM) patients and one pulmonary cryptococcosis (PC) patients were enrolled. The population included eight males and three females, with a mean age of 38.0±7.8 years (range 29–54 years). Among ten CM patients, the common clinical manifestation was headache, fever, vomiting and nausea. All of the ten CM patients were positive for CSF ink staining and nine of them were positive for CSF cryptococcal antigen latex agglutination test (titer 1:16–1:1024). Chest pain was the most obvious symptom in the patient with PC (GX0717). Chest CT showed a nodular shadow (0.6 cm in diameter) in the right lower lobe and multiple patchy shadow of the lungs; pathologic examination of the lung tissue biopsy demonstrated the cryptococcal yeast forms. Only one patient (GX0080) was previously diagnosed as systemic lupus erythematosus (SLE) and received corticosteroid treatment. Whereas, no significant immune abnormalities was detected in other patients.

**Table 1. Epidemiological and clinical characteristics of eleven patients with *Cryptococcus gattii* species complex infections in Guangxi, southern China\*.**

| Case no. | Age,y/ gender | Occupation | Clinical syndrome | Physical signs | Specimen cultured | Latex agglutination test | Clinical diagnosis | Treatment | Outcome |
|---|---|---|---|---|---|---|---|---|---|
| GX0049 | 37/M | Worker | Convulsion, altered consciousness, limb weakness, fever | Meningeal irritation and pathological reflex positive | CSF | 1:64 | CM | AMB(1275mg)+FLC(13.2g)+5FC (270g) | Survived |
| GX0079 | 40/M | Truck driver | Headache, vomiting, blurred vision | Neck stiffness | CSF | 1:256 | CM | Unknown | Survived |
| GX0080 | 31/F | Farmer | Fever, headache, nausea, vomiting, chills | Normal | CSF | 1:16 | CM | AMB(2035mg)+FLC(36.4g) | Survived |
| GX0158 | 54/F | Farmer | Headache, nausea, vomiting, fever | Normal | CSF | 1:128 | CM | AMB(2580mg)+FLC(22g)+5FC (299.5g) | Survived |
| GX1622 | 29/M | Farmer | Headache, vomiting, fever, blurred vision | Normal | CSF | 1:80 | CM | AMB(740mg)+FLC(8g) | Survived |
| GX0104 | 30/M | Farmer | Headache, nausea, vomiting, fever | Normal | CSF | 1:1024 | CM | AMB(1800mg)+FLC(39.6g)+5FC (52g) | Survived |
| GX0105 | 29/F | Unknown | Headache, fever, dizzy, chills | Neck stiffness, Kernig's sign positive | CSF | 1:64 | CM | AMB(1625mg)+FLC(26.4g)+5FC (248g) | Survived |
| GX0147 | 44/M | Farmer | Headache, fever, chills | Neck stiffness | CSF | 1:1024 | CM | AMB(189mg)+FLC(5g)+5FC (30g) | Died |
| GX0476 | 42/M | Unknown | Headache, fever, cough | Normal | CSF | None | CM | Unknown | Survived |
| GX0903 | 36/M | Aquiculture | Headache, dizzy, nausea, vomiting, fever | Neck stiffness | CSF | 1:32 | CM | AMB(2200mg)+FLC(20.8g)+5FC (278g) | Survived |
| GX0717 | 47/M | Civil servant | Chest pain, cough, expectoration | Normal | Lung tissue | None | PC | AMB(140mg)+FLC(2.4g) | Survived |

\*Cerebrospinal fluid, CSF; CM, Cryptococcal meningitis; PC, Pulmonary cryptococcosis

According to Infectious Diseases Society of America (IDSA) guidelines and China's expert consensus on the diagnosis and treatment of cryptococcal meningitis[36, 37], most of the patients have been significantly relieved after the cryptococcosis induction period treatment with combination therapy of two (AMB and FLC) or three (AMB, FLC and 5FC) antifungal drugs. Median cumulative dose and duration were 1625 mg (range 140–2580 mg) and 47 days (range 5–63 days) for AMB, 259 g (range 30–299.5 g) and 45.5 days (range 5–63 days) for 5FC, 20.8 g (range 2.4–39.6 g) and 50 days (range 6–66 days) for FLC. Even after the infection was treated and discharged from hospital, most patients continued the antifungal treatment with 0.2 g-0.4 g/day of FLC. Despite antifungal treatment, one patient died (GX0147) after giving up treatment because of no significant improvement. No patient had travel history outside China.

## Identification of sequence type

MLST analysis of the eleven isolates indicated that five isolates were *C. gattii s.s.*, and the other six isolates belonged to *C. deuterogattii* (Figs 1 and 2, Table 2). Five and three different sequence types (STs) were detected in *C. gattii s.s.* and *C. deuterogattii*, respectively; essentially all *C. gattii s.s.* isolates were genetically different and represented by a single isolate when MLST sequences were analyzed (Fig 1); sample GX0079 was quite divergent from the selected AFLP4/VGI strains. In contrast, *C. deuterogattii* isolates were divided into three major groups, with ST169 and ST129 represented by more than one isolate (Table 2).

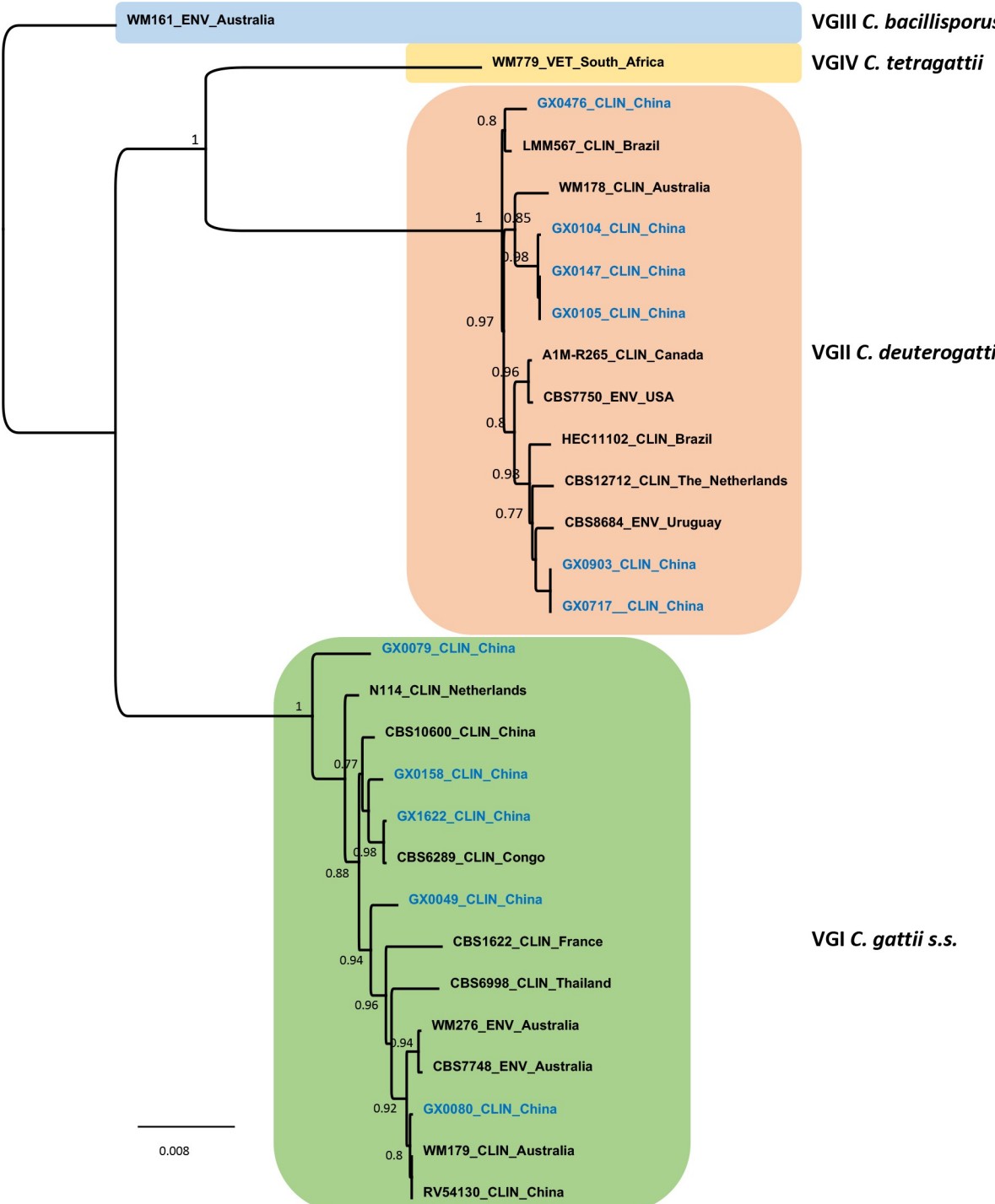

**Fig 1. Phylogenetic analysis of eleven *C. gattii* species complex.** Phylogenetic relationships inferred from a maximum likelihood analysis of *CAP59*, *GPD1*, *LAC1*, *IGS1*, *PLB1*, *SOD1* and *URA5* sequences of eleven *C. gattii* species complex from Guangxi, southern China (in blue) and 18 reference strains, covering the four major molecular types in *C. gattii* species complex. The branches with bootstrap support higher than 70% are shown.

Genetic analysis of the Chinese and global *C. deuterogattii* isolates indicated that four Chinese isolates (GX0104, GX0105, GX0147 and GX0476) clustered closely with the *C.*

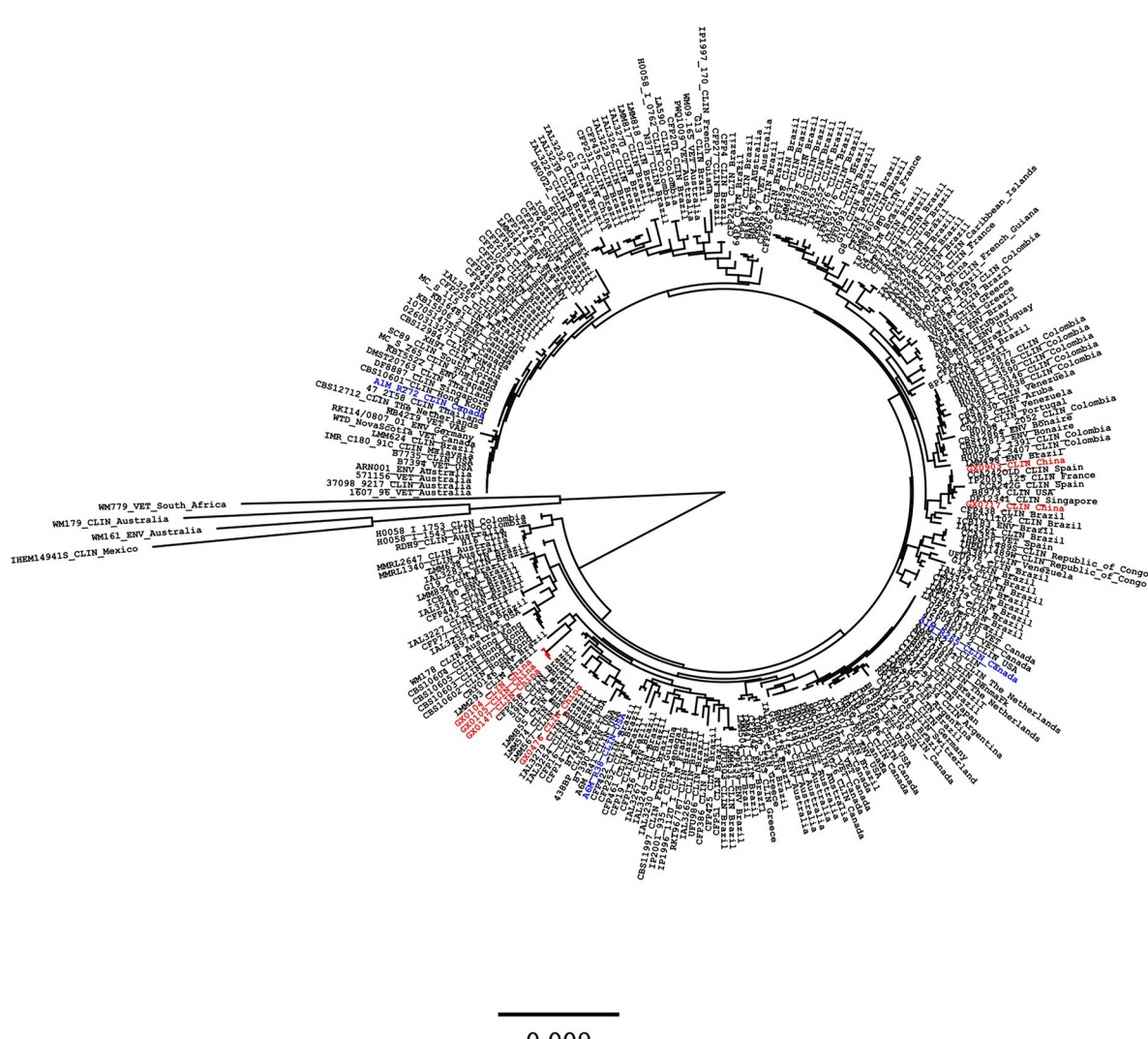

0.009

**Fig 2. Phylogenetic analysis of *C. deuterogattii* isolates.** The figure showing the relationships between the Chinese (strains of current investigation in red) and global *C. deuterogattii* strains inferred from the maximum likelihood analysis based on the combined ISHAM consensus MLST loci using MEGA7. Standard reference strains *C. gattii s.s.* (WM179), *C. bacillisporus* (WM161), *C. tetragattii* (WM779) and *C. decagattii* (IHEM14941) were used as outgroups. Epidemiologically significant isolates from the Pacific Northwest outbreaks were in blue. The branches with bootstrap support of more than 70% are indicated in bold.

*deuterogattii* AFLP6/VGII strains from Brazil (LMM293 and ILA3279), while the clinical isolates GX0903 and GX0717 clustered in a group containing *C. deuterogattii* AFLP6/VGII strains from Spain (CCA242OLD), Singapore (DF12341), the USA (B8973) and France (IP2003/125) (Fig 2). These isolates did not cluster with the highly virulent *C. deuterogattii* strains (AFLP6A/VGIIa R265 and AFLP6C/VGIIc A6M-R38) reported in the PNW outbreaks in Canada and the USA, but isolate GX0476 clustered in a group forming a sister relationship with the group harboring AFLP6C/VGIIc (A6M-R38) from Oregon, the USA.

## Comparison of Chinese and global isolates

PCA (based on ISHAM-MLST) was also used to assess the genetic relationship between Chinese and global isolates (S1 Fig). Approximately 41.8% of the genetic variation can be

**Table 2. The STs and mating types of clinical *Cryptococcus gattii* species complex isolates in Guangxi, southern China[*].**

| Strains | Allele type | | | | | | | ST | Molecular type | Mating type |
|---------|-------|------|------|------|------|------|------|-----|---------------|-------------|
|         | CAP59 | GPD1 | IGS1 | LAC1 | PLB1 | SOD1 | URA5 |     |               |             |
| GX0049 | 36 | 11 | 13 | 5  | 13 | 36  | 14 | 106 | VGI  | α |
| GX0079 | 49 | 11 | 59 | 13 | 13 | 71  | 24 | 227 | VGI  | α |
| GX0080 | 16 | 5  | 3  | 5  | 5  | 32  | 12 | 51  | VGI  | α |
| GX0158 | 36 | 11 | 83 | 13 | 13 | 47  | 15 | 222 | VGI  | α |
| GX1622 | 53 | 11 | 13 | 13 | 13 | 68  | 15 | 232 | VGI  | α |
| GX0104 | 40 | 35 | 57 | 4  | 1  | 59  | 2  | 129 | VGII | α |
| GX0105 | 8  | 6  | 25 | 4  | 1  | 16  | 6  | 129 | VGII | α |
| GX0147 | 40 | 35 | 57 | 4  | 1  | 59  | 2  | 129 | VGII | α |
| GX0476 | 2  | 35 | 57 | 4  | 1  | 104 | 2  | 309 | VGII | α |
| GX0903 | 2  | 25 | 26 | 21 | 9  | 8   | 7  | 169 | VGII | α |
| GX0717 | 2  | 25 | 26 | 21 | 9  | 8   | 7  | 169 | VGII | α |

[*]ST, sequence type

explained in *C. gattii s.s.* (PC1 30.0%, PC2 11.8%) (S1A Fig), and 35% of the variation was explained in *C. deuterogattii* (PC1 17.9%, PC2 17.1%) (S1B Fig). The PCA did not group *C. gattii* species complex strains according to their origin, which was consistent with the results of the phylogenetic analysis (S1 Fig). Among the *C. deuterogattii* isolates, the four *C. deuterogattii* isolates appeared to be originated from South America.

## Physiological characterization

All of the clinical isolates analyzed were *MAT*α (Fig 3A). Based on the crossing experiment with JEC20 (AFLP2/VNIV), three *C. deuterogattii* strains (GX0104, GX0105 and GX0147) were able to undergo sexual reproduction *in vitro*, and basidia and basidiospores were formed during sexual reproduction (Fig 3B). Two *C. deuterogattii* isolates (GX0105 and GX0147) exhibited a weak mating response, with only a few filaments, basidia and basidiospores developed.

The tested clinical isolates produced melanin at both 30°C and 37°C, while most strains had greater melanin production at 30°C than at 37°C (S2A Fig). Two *C. deuterogattii* strains (GX0104 and GX0903) appeared to have greater melanin production than the reference strain (AFLP6A/VGIIa R265) at 37°C. Moreover, these clinical strains produced capsule of different size *in vitro* (S2B Fig); however, our data revealed no significant difference ($p > 0.05$) in the capsule size between GX0104 and R265 (S2C Fig).

## Antifungal drug susceptibility

As the MICs shown in Table 3, none of the strains demonstrated resistance to the antifungal drugs. The susceptibility ranges of the five *C. gattii s.s.* isolates were 1–4 μg/mL for FLC, 0.125–0.5 μg/mL for 5FC, 0.25–0.5 μg/mL for AMB, 0.125–0.25 μg/mL for ITC, 0.0156–0.125 μg/mL for VOR, 0.0156–0.25 μg/mL for POS, and 0.0078–0.125 μg/mL for ISA. For the six *C. deuterogattii* isolates, the susceptibility ranges were 1–16 μg/mL for FLC, 0.0625–1 μg/mL for 5FC, 0.25–1 μg/mL for AMB, 0.0625–0.25 μg/mL for ITC, 0.0156–0.125 μg/mL for VOR, 0.0156–0.25 μg/mL for POS, and 0.0078–0.125 μg/mL for ISA. According to the clinical efficacy, combination of two or three drugs in AMB, FLC, and 5FC were effective for treatment. There were no significant differences between the *C. gattii s.s.* and *C. deuterogattii* isolates against these seven antifungal drugs ($p > 0.05$).

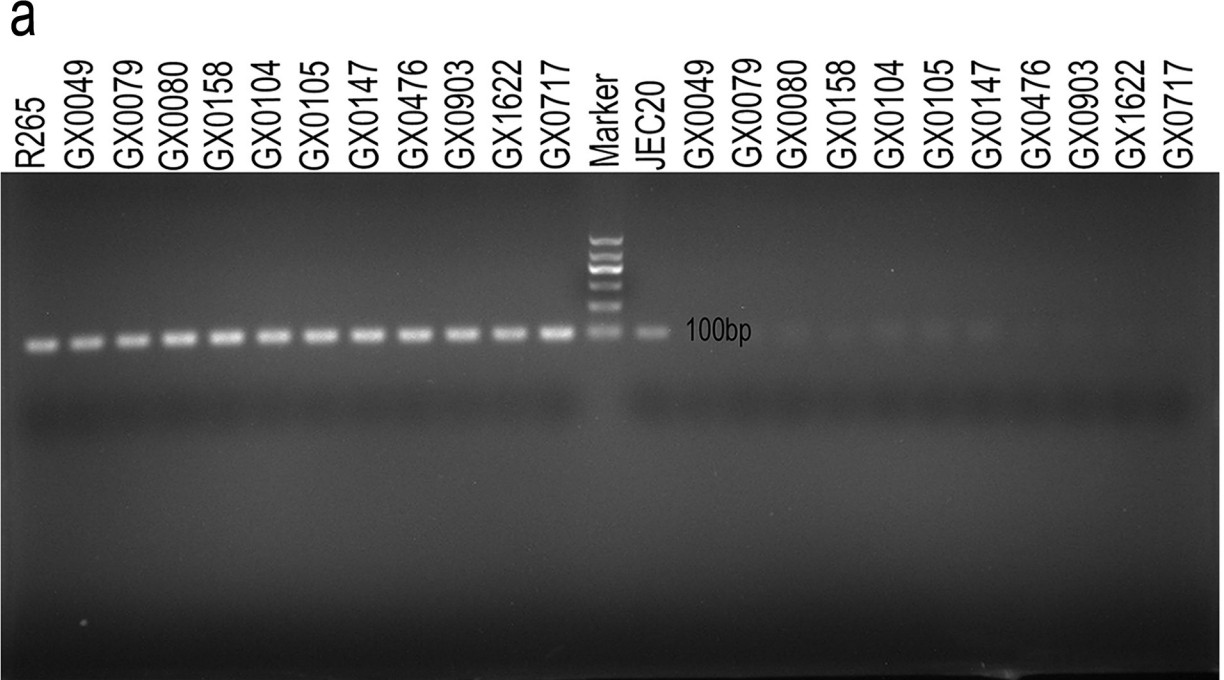

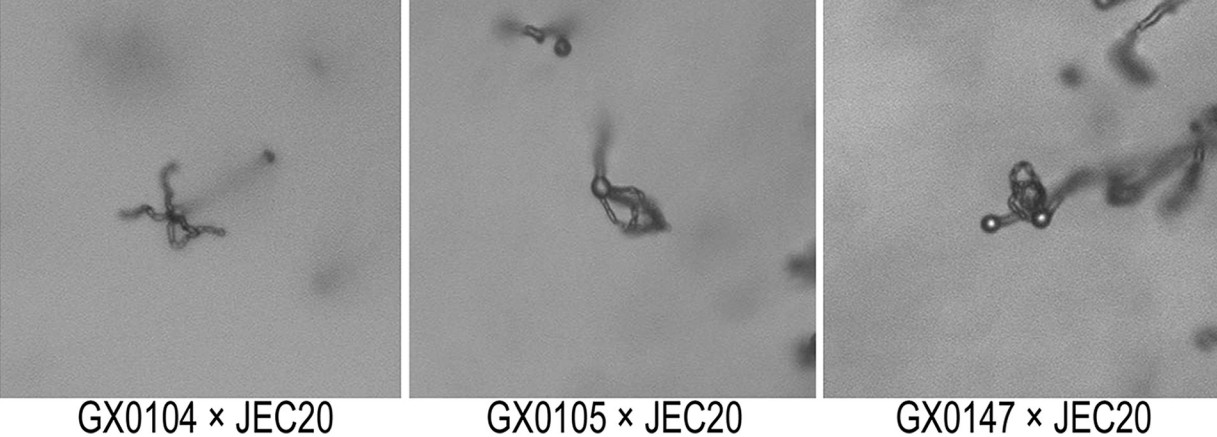

GX0104 × JEC20          GX0105 × JEC20          GX0147 × JEC20

**Fig 3. (A) Simultaneous amplification of all clinical strains and two reference strains using primers targeting the *MATα* and *MAT*a genes and (B) mating reactions of the isolates showing typical basidia and basidiospores. (A)** Reference strains: R265 (*MATα*, AFLP6A/VGIIa) and JEC20 (*MAT*a, AFLP2/VNIV). Clinical isolates were all *MATα*. **(B)** Sexual reproduction cultures including hyphae, basidia and basidiospores, incubated on V8 agar (pH = 5) at 25˚C for 2 weeks in the dark.

## Discussion

Our current study showed that *C. gattii* species complex infection in Guangxi, southern China were caused by both *C. gattii s.s.* (AFLP4/VGI) and *C. deuterogattii* (AFLP6/VGII), with different MLST genotypes and virulence factor characteristics. Similar pattern has also been reported in other countries such as Canada, Brazil, Australia and the USA[29–31]. Although *C. deuterogattii* in this study was genetically indistinguishable from the outbreak genotypes AFLP6A/VGIIa (R265) and AFLP6C/VGIIc (A6M-R38), they were related to other global *C.*

**Table 3. Minimal inhibitory concentrations (MICs) of all the *Cryptococcus gattii* species complex isolates in Guangxi, southern China\*.**

| Strains | Minimal inhibitory concentration (µg/mL) | | | | | | |
|---|---|---|---|---|---|---|---|
| | FLC | 5FC | AMB | ITC | VOR | POS | ISA |
| GX0049 | 1 | 0.5 | 0.25 | 0.125 | 0.0313 | 0.0625 | 0.0313 |
| GX0079 | 2 | 0.125 | 0.5 | 0.25 | 0.0625 | 0.0625 | 0.0313 |
| GX0080 | 1 | 0.25 | 0.5 | 0.25 | 0.0156 | 0.0156 | 0.0078 |
| GX0158 | 2 | 0.125 | 0.25 | 0.125 | 0.0313 | 0.0625 | 0.0313 |
| GX1622 | 4 | 0.5 | 0.5 | 0.25 | 0.125 | 0.25 | 0.125 |
| GX0104 | 1 | 0.0625 | 1 | 0.0625 | 0.0156 | 0.0156 | 0.0078 |
| GX0105 | 4 | 0.5 | 0.5 | 0.25 | 0.0625 | 0.0625 | 0.0625 |
| GX0147 | 4 | 1 | 0.5 | 0.25 | 0.0625 | 0.0625 | 0.0625 |
| GX0476 | 16 | 0.5 | 0.5 | 0.125 | 0.0625 | 0.25 | 0.125 |
| GX0903 | 2 | 0.25 | 0.25 | 0.25 | 0.0625 | 0.125 | 0.125 |
| GX0717 | 4 | 0.5 | 0.5 | 0.25 | 0.125 | 0.125 | 0.125 |
| MIC range | 1–16 | 0.125–1 | 0.25–1 | 0.0625–0.25 | 0.0156–0.125 | 0.0156–0.25 | 0.0078–0.125 |
| $MIC_{50}$ | 2 | 0.25 | 0.5 | 0.25 | 0.0625 | 0.0625 | 0.0625 |
| $MIC_{90}$ | 4 | 0.5 | 0.5 | 0.25 | 0.125 | 0.25 | 0.125 |
| Geometric mean | 2.5733 | 0.2836 | 0.4695 | 0.1824 | 0.0486 | 0.0709 | 0.0456 |

\* FLC, fluconazole; 5FC, fluorocytosine; AMB, amphotericin B; ITC, itraconazole; VOR, voriconazole; POS, posaconazole; ISA, isavuconazole; MIC, minimal inhibitory concentration

*deuterogattii* isolates isolated from Brazil (LMM293 and ILA3279), Singapore (DF12341) and Spain (CCA242OLD). *C. deuterogattii* strains are genetically diverse, and there was great genotypic variability among *C. deuterogattii* subtypes (AFLP6A/VGIIa and AFLP6C/VGIIc), particularly those in Brazil[38]. Despite the small sample size in this study, most Chinese clinical strains were genetically diverse, with eight STs reported. The variability could be due to special ecological interactions, adaptations and evolutionary mechanisms. *Cryptococcus* species are able to adapt to different ecological niches and temperatures and they can adapt within the host and environment through microevolution. The *Cryptococcus* genome is also dynamic in its plasticity[39].

In the present study, we found that all isolates harbored the *MAT*α gene at the mating type locus, and three of six *C. deuterogattii* strains (GX0104, GX0105 and GX0147) were able to undergo sexual reproduction, thus recombination could be one of the causes for the genetic diversity among the Chinese *C. deuterogattii* strains. Moreover, offspring produced by sexual reproduction/recombination may be able to adapt to and colonize new environment[40, 41]. In the mating assays of eleven clinical strains with the opposite mating type strains (AFLP6/VGII, AFLP2/VNIV and AFLP1/VNI), only the cross of AFLP6/VGII × AFLP2/VNIV exhibited mating response. Hybridization between the two cryptococcal species complexes seems to be a much rarer event in nature compared to hybridization between *C. neoformans* species complex lineages[42]. Even though the mating between the parental lineages can be induced in the laboratory, their progenies were not commonly reported in the environmental and clinical samples[43]. Recombination may also occur between the AFLP4/VGI and AFLP6/VGII isolates in China because previous research has highlighted possible gene transfer (introgression) between different *C. gattii s.l.* clades, either bisexually or unisexually, thereby contributing to the production of virulence subtypes[41, 44]. Sex can contribute to *de novo* diversity [45], which may be the reason for the diversity in the MLST analysis of clinical *C. deuterogattii* isolates. In addition, the capacity for mating could generate additional infectious propagules, leading to increased exposure and, ultimately, an enhanced infection rate[31].

Clinical data from this study showed that almost all patients had no obvious immunodeficiency, except one patient with SLE, which was consistent with the fact that the *C. gattii* species complex causes infection in immunocompetent hosts[11]. Most strains in Guangxi, southern China had the capabilities to produce melanin and capsule, as well as to undergo sexual reproduction. These are major virulence factors known to contribute to *Cryptococcus* pathogenesis and have been well characterized in *C. gattii* species complex. However, very few past studies have investigated these virulence factors in Chinese strains. Our data showed that most Chinese strains, similar to the reference outbreak strain R265 from Canada, have the abilities to be virulent. However, animal model studies need to be performed in order to establish their virulence levels to human hosts and whether their virulence would be comparable to the highly virulent outbreak (R265) and less virulent strain (R272) from the PNW[29].

Most patients significantly relieved after combination therapy while one patient (GX0147) passed away after giving up treatment because of no significant improvement. However, antifungal susceptibility testing indicated that strains in this study are not resistant to antifungal drugs because their MIC values were within the susceptible range[33–35]. Therefore, although none of the strains demonstrated resistance to these drugs, other factors may complicate the therapeutic outcomes of these patients.

*Cryptococcus* can be dispersed through the movement of trees and wood products, air currents, water currents, and biotic sources, such as birds, animals and insects[13]. Unlike *C. neoformans* species complex, which is found commonly in soils contaminated with wild and pet bird droppings, especially pigeon droppings, *C. gattii* species complex prefer to grow within and underneath moist bark, tree hollows, tree trunks and in soil debris near specific trees that are often found in nature (including *Eucalyptus*, *Azadirachta*, *Castanopsis*, *Prunus dulcis*, *Pinus canariensis* and *Pseudotsuga menziesii*)[46]. The evidence for *C. gattii* species complex dispersal by wind and air currents is limited, but fungal isolations from air samples have been obtained around positive trees in Canada and India[13]. "Guangxi Autonomous Region" belongs to southern China and lays on the southeastern corner of the Yunnan-Guizhou Plateau, situated from 20º54′N to 26º24′N and from 104º26′E to 112º04′E. This region borders Vietnam to the southwest and is surrounded by Guangdong, Guizhou, Yunnan, and Hunan Provinces in China. The region has a terraced topography sloping from the northwest to the southeast, with hilly land constituting 85% of its total area and plains constituting 15%. The region has a subtropical humid monsoon climate, with average daily temperatures of 16˚C to 23˚C. The rainy season lasts from April until September, with an annual rainfall of 1500 mm to 2000 mm[47]. The warm, subtropical monsoon climate in Guangxi is also favorable for the growth and reproduction of the *C. gattii* species complex. In this study, most patients were farmers in rural areas and had a history of contact with above plants and soil, especially *Eucalyptus*, but without travel record to the epidemic areas such as the PNW region, or the endemic area of *C. gattii* species complex such as South America and Australia, indicating that their cryptococcosis infection was spread through contact with the local environment. Frequent and regular contact with the natural environment may explain how these patients acquired the infections[48]. The genetic variabilities of Chinese *C. gattii* species complex members also suggested the possibility of multiple independent origins. Combined with the phylogenetic analysis of this study, some Chinese strains were related to those from Brazil. Therefore, this may be traced to the wood/seedlings imported from elsewhere, such as South America and Australia [49]. However, additional isolation and investigation of the *C. gattii* species complex from the environment such as trees, soil and wild animals is necessary to establish the environment as the source of infection in Guangxi Province, China.

Cryptococcosis infections caused by the *C. gattii* species complex are relatively rare in comparison to those caused by the *C. neoformans* species complex (11.4% versus 88.6%)[15];

however, recent studies have indicated that they may be mis- or underdiagnosed globally[50]. Similarly, the number of *C. gattii* species complex infections in China may be underestimated. Compared with the data from other provinces in China[14, 51], the rate of *C. gattii* species complex infection in Guangxi was approximately 9% (11/120), and most infection were caused by *C. deuterogattii*; however, additional epidemiological and surveillance studies are required.

There are several limitations of this study. First, a small number of patients were included in the study. Second, due to the retrospective nature of study, there was no long-term follow-up assessment of these patients, which have led to some missing data. Third, the level of virulence among these strains still needs further study on animal models. Fourth, environmental isolates should be sampled to assess the relationship of environmental niche for *C. gattii* species complex in this area. Thus, performing a prospective study with a larger study population are expected to elucidate the population structure and mechanisms of *C. gattii* species complex. Presently, we are in collaboration with other hospitals to carry out regional surveillance of *Cryptococcus* infections in China.

In summary, the *C. gattii* species complex should receive substantial attention in China due to its genetic variability, ability to infect immunocompetent hosts and propensity to undergo sexual reproduction and cause outbreaks, even the number of infections was low. Given that all patients in this investigation may have acquired the infection from nature, the environmental distribution, genetic variability and virulence level of the *C. gattii* species complex should not be underestimated. At the same time, we strive to improve the differential diagnosis of the *C. gattii* species complex in the early stages of infection and to use targeted treatment programs to reduce the risk of infection.

## Supporting information

**S1 Fig. Genetic relationships of *C. gattii s.s.*** (a) and *C. deuterogattii* (b) between Guangxi, southern China, and global isolates illustrated by principal component analysis (PCA). (PDF)

**S2 Fig.** (a)Visual analysis of melanin production after fungal growth on caffeic acid agar at 30˚C and 37˚C for three days. (b) Polysaccharide capsule surrounding the cells of *C. deuterogattii* isolates under microscopy. (c) Capsule production test in RPMI-1640 with 5% $CO_2$ at 37˚C representing the average capsule-capsule:cell wall-cell wall ratio of the six clinical isolates and the reference strain (AFLP6A/VGIIa R265). (X400, $p < 0.001$) (error bars ± SE = 2 SE). (PDF)

## Acknowledgments

We acknowledge all microbiologists at the First Affiliated Hospital of Guangxi Medical University for technical assistance and the isolation of *Cryptococcus* cultures. We also thank Ferry Hagen (Westerdijk Fungal Biodiversity Institute, the Netherlands) for discussion and sharing the sequence alignment of global *C. deuterogattii* (AFLP6/VGII) isolates.

## Author Contributions

**Conceptualization:** Chunyang Huang, Clement K. M. Tsui, Min Chen, Wanqing Liao, Cunwei Cao.

**Data curation:** Chunyang Huang, Clement K. M. Tsui.

**Formal analysis:** Chunyang Huang, Clement K. M. Tsui, Min Chen, Kaisu Pan, Meini Chen.

**Funding acquisition:** Cunwei Cao.

**Investigation:** Chunyang Huang, Kaisu Pan, Xiuying Li.

**Methodology:** Chunyang Huang, Kaisu Pan, Linqi Wang, Wanqing Liao, Cunwei Cao.

**Project administration:** Yanqing Zheng, Wanqing Liao, Cunwei Cao.

**Resources:** Chunyang Huang, Clement K. M. Tsui, Kaisu Pan, Xiuying Li, Linqi Wang, Dongyan Zheng, Xingchun Chen, Li Jiang, Lili Wei, Wanqing Liao, Cunwei Cao.

**Software:** Clement K. M. Tsui, Min Chen.

**Supervision:** Wanqing Liao, Cunwei Cao.

**Validation:** Chunyang Huang, Clement K. M. Tsui, Min Chen, Kaisu Pan.

**Visualization:** Chunyang Huang, Clement K. M. Tsui, Min Chen.

**Writing – original draft:** Chunyang Huang, Clement K. M. Tsui.

**Writing – review & editing:** Chunyang Huang, Clement K. M. Tsui, Min Chen, Wanqing Liao, Cunwei Cao.

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
