## [Decision Letter · Decision Letter 0]

10 Mar 2020

Dear Pro. cao,

Thank you very much for submitting your manuscript "Emerging Cryptococcus gattii species complex infection in south China" for consideration at PLOS Neglected Tropical Diseases. As with all papers reviewed by the journal, your manuscript was reviewed by members of the editorial board and by several independent reviewers. In light of the reviews (below this email), we would like to invite the resubmission of a significantly-revised version that takes into account the reviewers' comments. 

Additional editor comments: 120 clinical isolates are mentioned as the “population” from which these 11 strains were drawn. Were all consecutive cryptococcal isolates (cultured from patients at First Affiliated Hospital of Guangxi Medical University) stored for study purposes between 2014-2018? If not, please specify this. Provide information on the clinical source (specimen type) and patient characteristics for all 120 isolates, if this is possible. What does “the outer court treatment” in Table 1 refer to? Please summarise the main limitations of this study in the discussion section.

We cannot make any decision about publication until we have seen the revised manuscript and your response to the reviewers' comments. Your revised manuscript is also likely to be sent to reviewers for further evaluation.

Sincerely,

Nelesh P. Govender

Guest Editor

Todd Reynolds

Deputy Editor

Editor: 120 clinical isolates are mentioned as the “population” from which these 11 strains were drawn. Were all consecutive cryptococcal isolates (cultured from patients at First Affiliated Hospital of Guangxi Medical University) stored for study purposes between 2014-2018? If not, please specify this. Provide information on the clinical source (specimen type) and patient characteristics for all 120 isolates, if this is possible. What does “the outer court treatment” in Table 1 refer to? Please summarise the main limitations of this study in the discussion section.

Reviewer's Responses to Questions

**Key Review Criteria Required for Acceptance?**

**Methods**

-Are the objectives of the study clearly articulated with a clear testable hypothesis stated?

-Is the study design appropriate to address the stated objectives?

-Is the population clearly described and appropriate for the hypothesis being tested?

-Is the sample size sufficient to ensure adequate power to address the hypothesis being tested?

-Were correct statistical analysis used to support conclusions?

-Are there concerns about ethical or regulatory requirements being met?

Reviewer #1: (No Response)

Reviewer #2: (No Response)

Reviewer #3: The objectives are clear with the appropriate design. The screening of 120 clinical strains between 2014-2018 by L-concanavanine-glycine-bromthymol blue (CGB) medium showed 11 Cryptococcus gattii species complex from Non-HIV patients in South China. Five of C. gattii ss (and six of C. deuterogattii (AFLP6A/VGIIa) Mating and susceptibility test against 7 common antifungal agents were investigated. The genetic variation and phylogenetic tree using MLST information were analyzed fertility

However:

1. The first sentence of materials and methods: 120 clinical strains were collected....... It is not clear whether 120 clinical strains of which group of organisms they recruited for CGB screening.

2. The demography, association diseases, antifungal agents profiles, genetic variation, and phylogenetic trees were analyzed based on 11 isolates after screening of 120 isolates (might be suspected Cryptococcus spp.??).

**Results**

-Does the analysis presented match the analysis plan?

-Are the results clearly and completely presented?

-Are the figures (Tables, Images) of sufficient quality for clarity?

Reviewer #1: (No Response)

Reviewer #2: (No Response)

Reviewer #3: - Table 1: -- More detail of clinical syndrome and temp. of fever should be mentioned to let the readers learn from experience of the authors and any other laboratory diagnosis was examined or not, ie: cryptococcal antigen. 

-- Environment contact : Eucalyptus and pigeon. Actually Cryptococcus's habitat is not only the Eucalyptus and pigeon. It can be the tree bark, the hollow of big trees and other avian exclude from pigeon. Thus, isn't it leads the readers to misunderstand the environment niche. 

- Genetic variation and phylogenetic trees based on the 7 loci-MSLT performing were analyzed and the result showed the association with the global isolates.

- Fertility was detected. Capsule sIze and melanin pigment formation were all comparable to standard strains.

- The 7 antifungal agents profiles against these 11 C. gattii cpx according to the CLSI protocol were investigated.

- The antifungal drug dosage and duration of treatment are other significance factors to let the readers get some information for the survivals and decease.

**Conclusions**

-Are the conclusions supported by the data presented?

-Are the limitations of analysis clearly described?

-Do the authors discuss how these data can be helpful to advance our understanding of the topic under study?

-Is public health relevance addressed?

Reviewer #1: (No Response)

Reviewer #2: (No Response)

Reviewer #3: This authors concluded that the mating α C. gattii complex were isolated from clinical specimens in South China between 2014-2018 and all these emerging 11 isolates, (10%, if 120 isolates were Cryptococcus spp.) are all susceptible to antifungal agents. The small genetic variation, 8STs, in their isolates were found and related to the reported strains from other countries. The distribution of C. gattii cpx. in this area might due to the capability of sexual reproduction. The authors proposed the future study to confirm the virulence in animal model.

**Editorial and Data Presentation Modifications?**

Reviewer #1: (No Response)

Reviewer #2: (No Response)

Reviewer #3: Minor revision: - as mentioned above regarding more clinical symptom, antifungal dosage,

- the original group in the CGB screening, more explanation about the environment niche, not only Eucalyptus tree,..

- Clinical manifestation and/or diagnosis should provide more information, ie: GX0717 chest pain and specimens : Lung tissue. This information does not provide any more info. It is just like common things. How the X-ray? How is

- The best condition for Laccase gene: melanin pigment formation will be 30C is already the theory so I donot think this will give any benefit for this experiment. 

-The limitation of sample should be focused in discussion.

**Summary and General Comments**

Reviewer #1: Dear Editor

The manuscript entitled “Emerging Cryptococcus gattii species complex infection in south China” by Huang et al, presented the link between C. deuterogattii isolates from south China and those from Brazil, Spain, Singapore, USA and France. Although the number of isolates in this study was quite small (N=11), this report would trigger some interesting points for further investigation to see how C. gattii complex spreads worldwide. There are some comments for the manuscript.

1. For the mating experiment, why the authors performed the experiment using JEC20 to mate with C. gattii complex. The JEC20 is C. neoformans serotype D, not C. gattii complex. The authors should use C. gattii complex MATa to do the experiment to make it more convincing. In the discussion Line 318-322, the authors mentioned that sexual reproduction could cause genetic diversity from these results. Is it usual that C. gattii mate with C. neoformans in the environment? 

2. The capsule and melanin production in these virulent strains is not unexpected as those virulent factors are well-known. Therefore, these data did not add anything new. To make it more convincing, the capsule and melanin experiments need a non-virulent (environmental) C. gattii strain as a negative control (in addition to R265 strain). In this case, R272 (environmental C. gattii isolate) may be used as a control to show that it produces less capsule and melanin that the China isolates.

3. Line284 “……our data revealed no significant different (p < 0.01) in ….”. This is confusing why it was not significant as the p value was < 0.01. 

4. The authors should discuss why they use triple drug combination to treat C. gattii infection (amphotericin B plus fluconazole plus flucytosine). Is there any guideline in China recommend this combination therapy?

Reviewer #2: (No Response)

Reviewer #3: The authors tried to provide the emerging information of C. gattii complex in 2014-2018 in South China and demonstrated how significant they are thru the virulence factors, mating, capsule formation, melanin formation. In addition, the genetic variation of their strains are associate with the global strains which imply that this is not the result from recombination inside South China.

PLOS authors have the option to publish the peer review history of their article (what does this mean?). If published, this will include your full peer review and any attached files.

Reviewer #1: No

Reviewer #2: No

Reviewer #3: No
---

## [Decision Letter · Decision Letter 1]

17 Jun 2020

Dear Pro. cao,

We are pleased to inform you that your manuscript 'Emerging Cryptococcus gattii species complex infection in Guangxi, southern China' has been provisionally accepted for publication in PLOS Neglected Tropical Diseases.

Before your manuscript can be formally accepted, you will need to complete some formatting changes, but you will  will receive information on these formatting changes in a follow up email. A member of our team will be in touch with a set of requests.

Best regards,

Todd B. Reynolds

Deputy Editor

Todd Reynolds

Deputy Editor

Reviewer's Responses to Questions

**Key Review Criteria Required for Acceptance?**

**Methods**

-Are the objectives of the study clearly articulated with a clear testable hypothesis stated?

-Is the study design appropriate to address the stated objectives?

-Is the population clearly described and appropriate for the hypothesis being tested?

-Is the sample size sufficient to ensure adequate power to address the hypothesis being tested?

-Were correct statistical analysis used to support conclusions?

-Are there concerns about ethical or regulatory requirements being met?

Reviewer #1: (No Response)

Reviewer #2: (No Response)

Reviewer #3: The content in materials and methods are corresponded to the listed questions except the grammar of the verb after the data shouldn't be "were" not was (L173). The authors revised and provided the clear image of how the isolates derived from and how to proof the isolates which the experiments used the standard methods. Multilocus sequencing typing (MLST), one of the standard and verified tool to analyse the new sequence types were performed . However, the authors should provide the full name before write the short name (L.196). The ethical concern was clarified.

**Results**

-Does the analysis presented match the analysis plan?

-Are the results clearly and completely presented?

-Are the figures (Tables, Images) of sufficient quality for clarity?

Reviewer #1: (No Response)

Reviewer #2: (No Response)

Reviewer #3: The authors edited the results as comments, making the clear results. Each issue was clarified and finally the authors could show the new STs and characterised Cryptococcus gattii in Guangxi, southern China and showed how these eleven isolates related to Cryptococcus in other locations. Their antifungal profiles and demography of patients were laborated.

**Conclusions**

-Are the conclusions supported by the data presented?

-Are the limitations of analysis clearly described?

-Do the authors discuss how these data can be helpful to advance our understanding of the topic under study?

-Is public health relevance addressed?

Reviewer #1: (No Response)

Reviewer #2: (No Response)

Reviewer #3: The summary and the limitations were explained.

**Editorial and Data Presentation Modifications?**

Reviewer #1: (No Response)

Reviewer #2: (No Response)

Reviewer #3: Two points were already pointed out: grammatical error and the full name.

**Summary and General Comments**

Reviewer #1: Dear editor,

The authors have explained clearly in the "response to reviewers" and the revised manuscript. The data they presented would interest the readers. I have no further question.

Regards,

Reviewer #2: (No Response)

Reviewer #3: (No Response)

PLOS authors have the option to publish the peer review history of their article (what does this mean?). If published, this will include your full peer review and any attached files.

Reviewer #1: No

Reviewer #2: No

Reviewer #3: No

---

## [Editor Report · Acceptance letter]

10 Aug 2020

Dear Pro. cao,

We are delighted to inform you that your manuscript, "Emerging Cryptococcus gattii species complex infections in Guangxi, southern China," has been formally accepted for publication in PLOS Neglected Tropical Diseases.

Best regards,

Shaden Kamhawi

co-Editor-in-Chief

Paul Brindley

co-Editor-in-Chief
